# Association of Dietary Fish and n-3 Unsaturated Fatty Acid Consumption with Diabetic Nephropathy from a District Hospital in Northern Taiwan

**DOI:** 10.3390/nu14102148

**Published:** 2022-05-21

**Authors:** Shih-Ping Lin, Chiao-Ming Chen, Kang-Ling Wang, Kun-Lin Wu, Sing-Chung Li

**Affiliations:** 1School of Nutrition and Health Sciences, College of Nutrition, Taipei Medical University, Taipei City 11031, Taiwan; ping06072008@gmail.com; 2Department of Dietetics, Taoyuan Armed Forces General Hospital, Taoyuan, Taoyuan City 32551, Taiwan; 3Department of Food Science, Nutrition, and Nutraceutical Biotechnology, Shih Chien University, Taipei City 10462, Taiwan; charming@g2.usc.edu.tw; 4General Clinical Research Center, Taipei Veterans General Hospital, Taipei City 11217, Taiwan; klwang@aftygh.gov.tw; 5Department of Medical Research and Education, Taipei Veterans General Hospital, Taipei City 11217, Taiwan; 6Department of Medicine, Taipei Veterans General Hospital, Taipei City 11217, Taiwan; 7Division of Nephrology, Department of Medicine, Tri-Service General Hospital, National Defense Medical Center, Taipei City 11031, Taiwan; ndmc6217316@aftygh.gov.tw; 8Department of Medicine, Taoyuan Armed Forces General Hospital, Taoyuan City 32551, Taiwan

**Keywords:** diabetes mellitus, diabetic nephropathy, urinary albumin–creatinine ratio, estimated glomerular filtration rate, semiquantitative food frequency questionnaire

## Abstract

Nephropathy caused by diabetes mellitus (DM) is the main cause of end-stage renal disease (ESRD). To understand the association of dietary intake with renal function indicators among patients with diabetic nephropathy (DN), this cross-sectional study was conducted at the dietetic consultation clinic of the Taoyuan Armed Forces General Hospital in Taiwan. In total, 317 participants were recruited for this study. Patients with diabetes who had a urinary albumin–creatinine ratio (UACR) of ≥30 mg/g were defined as having DN. The anthropometric characteristics, blood biochemistry, and renal function of the participants were assessed. Furthermore, a semiquantitative food frequency questionnaire (SQFFQ) was administered to investigate the dietary intake of the participants in the DM and DN groups. The result showed that participants in the DN group were older, had longer diabetes duration and poorer glycemic control and renal function than those in the DM group. Logistic regression models revealed that intake of high-fat marine fishes had the lowest odds ratio (OR) for DN risk compared with other fishes (OR: 0.868; 95% CI: 0.781–0.965, *p* = 0.009). Shellfish, soybean products, and skim milk also provided better protective effects to decrease the risk of DN. A further analysis of polyunsaturated fatty acids revealed that Σn-3 PUFAs significantly reduced DN risk, while Σn-6 PUFAs did not, especially EPA (OR: 0.821; 95% CI: 0.688–0.979, *p* = 0.029) and DHA (OR: 0.903; 95% CI: 0.823–0.992, *p* = 0.033) regardless of whether the variables were adjusted, including diabetes duration, age, and HbA1c. Our findings suggest that a diet that incorporates high-fat fish, shellfish, soybean products, and a lower Σn-6/Σn-3 ratio can mitigate DN risk.

## 1. Introduction

The prevalence of diabetes mellitus (DM) is rising annually. The 10th edition of the International Diabetes Federation Diabetes Atlas estimated that the proportion of patients with DM worldwide, which was 10.5% (536.6 million people) in 2021, will rise to 12.2% (783.2 million people) by 2045; this is equivalent to a 46% rising in the DM population [1]. Diabetic kidney disease (DKD) and diabetic nephropathy (DN) are the most common causes of chronic kidney disease (CKD) and the leading cause of end-stage renal disease (ESRD) in the United States and most developed countries, and 30–40% of patients with DM develop DN [2,3]. The clinical symptoms of DN include an abnormal urinary albumin excretion level, which is defined as microalbuminuria or albuminuria (i.e., spot urine albumin–creatinine ratio (UACR) ≥ 30 mg/g, or ≥300 mg/g, respectively) and is used as a diagnostic threshold for assessing renal damage [4,5]. Epidemiologic studies have suggested that patients with both ESRD and DM are at an increased risk of cardiovascular diseases and have higher mortality than those with only DM [6,7].

The incidence of renal dialysis in Taiwan increased from 10,177 in 2010 to 10,663 in 2014. Among patients, the most common comorbidities were hypertension, diabetes, and cardiovascular disease, and their prevalence rates were 89.2%, 62.5%, and 52.5%, respectively [8]. Furthermore, the 2015 annual report of the United States Renal Data System showed that among various countries, the highest incidences of treated ESRD were reported by Taiwan, Mexico, and the United States (458, 421, and 363 per million population, respectively) [9]. Undoubtedly, Taiwan had the world’s highest incidence and prevalence of ESRD in 2018. Hyperglycemia, hypertension, and hereditary factors are the main risk factors for developing DN. In addition, glomerular hyperfiltration, proteinuria level, smoking, and the quality and quantity of the protein and fat in a diet are also crucial factors that influence the development of DN [10].

Optimizing diet and nutrition is an important and modifiable factor that may prevent or delay the development of DN. Specifically, such patients should avoid the excessive consumption of carbohydrate- and protein-rich foods while favoring the inclusion of essential fats in their diets [11]. Although the long-term excessive consumption of fat contributes to body fat accumulation and obesity, which increases the risk of type 2 diabetes (T2DM) and cardiovascular diseases (CVDs), numerous studies have suggested that patients with CKD should consume high-fat diets with enough calories to compensate for caloric losses due to decreased protein intake. However, these patients should consume the appropriate types and amounts of fat to prevent fat accumulation in the body [12,13]. The National Kidney Foundation–Kidney Disease Outcomes Quality Initiative guidelines were published in 2021, and they emphasize that CVD prevention is the top priority for patients with CKD who exhibit dyslipidemia [14]. A diet that promotes CVD prevention is rich in fruits, vegetables, legumes, nuts, seeds, plant protein, and fatty fish; it should also contain less saturated fat, dietary cholesterol, salt, and refined grains [15].

Dietary fish oil supplementation prevents the progression of renal disease in patients with IgA nephropathy [16]. Han et al. reported that patients with both DM and hypertriglyceridemia who supplemented 4 g of n-3 polyunsaturated fatty acid (PUFA) daily achieved significantly favorable outcomes in maintaining their renal function compared with those who consumed lower dosages (1 g/day, 2 g/day) in a retrospective study; they also reported that this effect was dose-dependent [17]. A nested case-control study involving young Swedish patients with type 1 DM revealed an association between a higher intake of fish protein and a lower risk of microalbuminuria [18]. A prospective study conducted by Kutner et al. revealed a relationship between increased fish consumption and a reduction in mortality (50%) among a cohort of 216 patients on dialysis over a 3-year period [19]. Friedman et al. conducted a retrospective study. They measured baseline erythrocyte n-3 PUFA levels of hemodialysis patients and summed the erythrocyte EPA and DHA content as an omega-3 index (O3I). In a multivariate model, a protective trend was observed with an O3I above the median (Hazard Ratio (95% CI); 2.48 (0.88, 6.95)) [20].

Studies have suggested that patients with DN can delay the progression of the disease and reduce urinary protein excretion by increasing their consumption of fish and long-chain n-3 PUFA. Such surveys are relatively rare in Taiwan. In the present study, we conducted a cross-sectional study and recruited T2DM patients from a district hospital in Northern Taiwan. Dietary intake assessment was administered by an interviewer-administered food frequency questionnaire. The aim of this study was to investigate the association of fish and n-3 unsaturated fatty acid consumption with DN. The findings of this study can serve as a reference for formulating nutritional care plans and recommended dietary allowance guidelines for patients with DN.

## 2. Materials and Methods

### 2.1. Study Participants

The participants of the present study were recruited from the dietetic consultation clinic of the Taoyuan Armed Forces General Hospital in Taiwan. Patients were recruited if they were (1) diagnosed with T2DM at an endocrinology and metabolism clinic at least 1 year prior to the start of the study and adhered to the treatments prescribed by their physicians, (2) aged between 30 and 85 years, (3) and following their physician’s advice regarding oral medication and insulin administration. Patients with immune-related or autoimmune diseases, drug addiction, malignant tumors, cirrhosis, coagulation abnormality, or gallbladder diseases; patients on dialysis for ESRD; and patients who were being treated with oral coagulants were excluded from the present study.

All participants were fully informed of the procedures, objectives of the present study, and the content of the administered questionnaires, and then signed informed consent. All the participants with T2DM who were recruited for the first clinical study (June 2012 to May 2013) were approved by the Joint Institutional Review Board of Taipei Medical University (No. 201207004), and those who were recruited for the second clinical study (January to December 2021) were approved by the Institutional Review Board of Tri-Service General Hospital, National Defense Medical Center (No. 202005174).

### 2.2. Research Design

A cross-sectional study was conducted to collect the data from participants with diabetes, including basic characteristics, anthropometrics, laboratory measurements, and dietary intake assessments. Basic characteristics (i.e., sex, age, duration of diabetes, and prescription drug use) were collected from medical records of Taoyuan Armed Forces General Hospital; anthropometric and laboratory data were measured by medical professionals on a visit. Dietary questionnaires were also administered to observe the effect of diet on DN. Before the start of the experiment, each participant was interviewed and asked to comply with the instructions of outpatient dietitians; they were also informed of the content of the questionnaire and consent form, the objectives and procedures of the present study, and the requirements during the study period. In total, 317 participants were recruited for the present study. However, among the 317 participants, four who did not complete the administered questionnaires or did not meet the recruitment criteria were subsequently excluded. During the experiment, the participants maintain their normal life without any restrictions and continued their treatments as prescribed by their physicians.

### 2.3. Dietary Intake Assessment

An interviewer-administered semiquantitative food frequency questionnaire (SQFFQ) was administered to collect data on the dietary intake of protein (from various sources) and cooking oil. The protein-containing food items listed in the questionnaire were foods that are commonly consumed in Taiwan, such as eggs (e.g., chicken, duck, and salted eggs), meats (e.g., chicken, duck, pork, beef, and offal), low-fat marine fish (e.g., perch, hairtail, tuna, and butterfish), moderate-fat marine fish (e.g., narrowbarred mackerel, spotted mackerel, grouper, and mullet), high-fat marine fish (e.g., milkfish, salmon, saury, codfish, oilfish, and mackerel), freshwater fish (e.g., tilapia, yellow croaker, and grass carp), processed fish products (e.g., dried small fish, tempura, and fish balls), shellfish (e.g., shrimp, crab, lobster, clams, oysters, abalone, and babylon shell), dairy products (e.g., whole, low-fat, and skim milk), soybean products (e.g., tofu, dried bean dried tofu, soy milk, and vegetarian chicken), and seeds and nuts (e.g., pistachios, almonds, cashew nuts, flax seeds, peanuts, sunflower seeds, and pumpkin seeds). The consumption of edible oils was also assessed in the questionnaire; such oils included animal fat, soybean oil, sunflower oil, olive oil, peanut oil, rapeseed oil, sesame oil, and grapeseed oil. The content of the questionnaire was validated through an expert review. Two professors were invited to conduct a validity test of the questionnaire content. They examined the relevance and clarity of the content and assessed the usability of the questionnaire. The eligible participants were interviewed by a registered dietitian who then completed the SQFFQ using various food models. The frequency of food consumption within a single month was classified according to the six levels as follows: (1) no intake, (2) <1 time/week, (3) 1–3 times/week, (4) 4–6 times/week, (5) daily, and (6) ≥2 times/day. We converted servings of each type of food into weight and analyzed fatty acid composition using the government-published Taiwan Food Nutrition Database 2020 [21]. Σn-6 PUFA was the total amount of C18:2n-6 and C20:4n-6; Σn-3 PUFA was the total amount of C18:3n-3, C20:5n-3 (eicosapentaenoic acid, EPA), and C22:6n-3 (docosahexaenoic acid, DHA); MUFA was the total amount of C18:1n-9.

### 2.4. Anthropometry and Blood Biochemical Analysis

Body height and weight were measured using a height and weight measuring device. The body mass index (BMI) of a participant was calculated by dividing their weight (kg) by the square of their height (m). After a participant had rested for ≥10 min in a quiet room, their blood pressure was measured twice with a 10 min rest between measurements by using an FT-500 R automatic blood pressure meter (Jawon Medical, South Korea). The final blood pressure value was the mean of two measurements. Waist circumference was measured by using a plastic measuring tape at the mid-point between the iliac crest (hip bone) and costal margin (lowest rib).

Blood biomarkers, including fasting plasma glucose, creatinine, total cholesterol, and triglyceride were measured by enzymatic colorimetric assay using a Cobas c 501 analyzer (Roche Diagnostics, Mannheim, Germany). Estimated glomerular filtration rate (eGFR) was calculated using a modified Modification of Diet in Renal Disease formula, which is as follows: eGFR (mL/min/1.73 m^2^) = 175 × (creatinine, mg/dL)^−1.154^ × (Age, years)^−0.203^ × 0.742 (if female). Glycated hemoglobin (HbA1c) was analyzed through high-performance liquid chromatography (Spotchem Sp 4410, Arkary, Kyoto, Japan). The quantitative estimation of urinary protein was performed by applying the colorimetric sulfosalicylic acid method. The colorimetric estimation of urinary creatinine was performed by applying the modified Jaffe’s method [22].

### 2.5. Statistical Analysis

In the first clinical trial, we did not have enough funds, so we only collected 106 cases. In 2021, we received funding from Taoyuan Armed Forces General Hospital (grant number: TYAFGH-A-110006), so we conducted a second clinical trial and collected another 211 cases. The two clinical trials had consistent results, so the data were combined for statistical analysis.

Patients with diabetes who had a urinary albumin–creatinine ratio (UACR) of ≥30 mg/g were defined as having DN and those with UACR < 30 mg/g were defined as DM [23]. We divided the participants into two groups on the basis of this definition, namely, the DN and DM groups. All data were verified to have a normal distribution through the Kolmogorov–Smirnov test.

The data pertaining to basic demographic information, anthropometry, biochemical analysis, urinalysis, weekly food consumption, and dietary fatty acid composition were presented as means ± standard deviations (SDs) or percentage values. Intergroup differences were identified through independent *t* tests. Pearson’s chi-square test was used to assess categorical variables. Correlations were identified by performing multivariable logistic regressions to assess the associations between selected food consumption or dietary fatty acids and DN risk. There were no covariates were adjusted in Model 1; the diabetes duration, age, and HbA1c were adjusted in Model 2. All data analyses were performed using SPSS (version 19; SPSS, Chicago, IL, USA). A significant difference was recognized when *p* < 0.05.

## 3. Results

### 3.1. Participant Characteristics

We recruited 106 and 211 participants from Taoyuan Armed Forces General Hospital in 2012 and 2021, respectively. Four participants who did not meet the inclusion criteria or provided incomplete data were excluded. Of the 313 remaining participants, 162 were men, and 151 were women. They were divided into the DM (n = 169, 54%) and DN (n = 144, 46%) groups on the basis of their UACR measurements. The flow of the participant recruitment and assignment process is illustrated in Figure 1.

Table 1 presents the participants’ demographic information and their results with respect to their anthropometry, medication history, and hematological and urinary analysis. The participants were aged between 40 and 80 years. No significant difference between the DM and DN groups was detected for sex, height, weight, BMI, waist circumstance, blood pressure, and cholesterol level. However, compared to the participants in the DM group, the DN group were older (66.0 ± 11.1 vs. 62.7 ± 12.5), longer duration of diabetes (12.8 ± 8.6 vs. 9.3 ± 7.4 years), and poorer glycemic control (HbA1c; 7.9 ± 1.4 vs. 7.3 ± 1.2%) and renal function (UACR: 349.9 ± 692.6 vs. 12.2 ± 8.0 mg/g; eGFR: 72.7 ± 30.7 vs. 85.6 ± 22.8 mg/dL). We also analyzed the oral hypoglycemic agents used in participants, including sulfonylurea, biguanide, α-glucosidase inhibitor, SGLT2 inhibitor, glucagon-like peptide 1, thiazolidinedione, and dipeptidyl peptidase-4 inhibitor. There was no statistical difference in drug class between the groups, except that the DN group had a higher proportion of SGLT2 inhibitor than the DM group (data not shown). We noticed that the DN group took significantly more types of drugs for glycemic control than the DM group. These findings implied that participants in the DN group had more difficulty in controlling blood sugar than those in the DM group. There was no significant difference in antihypertensive drugs and insulin injections between the two groups.

### 3.2. Consumption of Selected Foods as Assessed through SQFFQ

The dietary records of total energy intake per day were 1726.2 ± 334.8 kcal in the DM and 1714.1 ± 776.9 kcal the in DN group. There was no difference significant difference between groups. Table 2 presents the results for the consumption of selected foods (number of servings per week) by the DM and DN groups. No significant difference between the two groups was detected for the following food items: eggs, meat, offal, marine fish (low- and moderate-fat marine fish), freshwater fish, processed fish products, dairy products, fats and oils, and nuts and seeds. Relative to the DM group, the DN group had a significantly lower intake of the following food items: high-fat marine fish (2.3 ± 2.3 (DN) vs. 3.0 ± 2.3 (DM)), shellfish (0.5 ± 0.9 (DN) vs. 0.7 ± 1.2 (DM)), soybean products (1.9 ± 2.0 (DN) vs. 2.5 ± 2.6 (DM)), olive oil (2.3 ± 3.6 (DN) vs. 3.1 ± 3.9 (DM)), and other oils (0.7 ± 2.3 (DN) vs. 1.4 ± 2.9 (DM)).

### 3.3. Association between Selected Foods Consumption and DN

Logistic regression models were used to examine the associations between DN risk and the consumption of selected foods. In Model 1, no covariates were adjusted; in Model 2, diabetes duration, age, and HbA1c were adjusted. Our data revealed that high-fat marine fish intake had the lowest odds ratio (OR) of DN risk compared with other fishes (OR: 0.876; 95% CI: 0.792–0.969). Shellfish, soybean products, and skim milk also provided a better protective effect to decrease the risk of DN (OR: 0.709; 95% CI: 0.554–0.908, OR: 0.894; 95% CI: 0.800–0.998, and OR: 0.561; 95% CI: 0.319–0.988, respectively) after adjusting for covariates (Table 3). Although olive oil decreased the risk of DN (OR: 0.928; 95% CI: 0.875–0.985) in Model 1 (*p* = 0.014), there was no statistical difference after adjusting for covariates (*p* = 0.062).

### 3.4. Dietary Fatty Acid Composition in DM and DN Groups

Dietary fatty acids composition in DM and DN groups were shown in Table 4. Relative to the DM group, the DN group had a significantly lower mean intake for the following fatty acids: total saturated fatty acids (30.1 ± 13.1 g (DN) vs. 33.8 ± 13.3 g (DM)), total monounsaturated fatty acids (39.6 ± 15.8 g (DN) vs. 44.1 ± 15.7 g (DM)), and total polyunsaturated fatty acids (28.6 ± 11.0 g (DN) vs. 31.5 ± 10.8 g (DM)). A further analysis of polyunsaturated fatty acids revealed that the intake of Σn-3, eicosapentaenoic acid (EPA), and docosahexaenoic acid (DHA) was lower in the DN group than in the DM group, such that the DN group had a higher Σn-6/Σn-3 ratio.

### 3.5. Association between Dietary Fatty Acids and DN

Logistic regression models were employed to examine the associations between DN risk and selected dietary fatty acids. In Model 1, no covariates were adjusted; in Model 2, diabetes duration, age, and HbA1c were adjusted. Our data revealed that total saturated fatty acids, total monounsaturated fatty acids, total polyunsaturated fatty acids, Σn-3, EPA, DHA, and the Σn-6/Σn-3 ratio were major indicators for predicting DN risk (Table 5). We noted that Σn-3 PUFAs significantly reduced DN risk, while Σn-6 PUFAs did not, especially EPA (OR: 0.821; 95% CI: 0.688–0.979, *p* = 0.029) and DHA (OR: 0.903; 95% CI: 0.823–0.992, *p* = 0.033) regardless of whether the variables were adjusted. In addition, a higher in the Σn-6/Σn-3 ratio also increased the risk of DN (OR: 1.088; 95% CI: 1.010–1.172, *p* = 0.027).

## 4. Discussion

Our result showed that participants in the DN group were older, had longer diabetes duration and poorer glycemic control and renal function than those in the DM group. Consumption of high-fat marine fish, shellfish, soybean products, and skim milk provided a better protective effect to decrease the risk of DN. A further analysis of polyunsaturated fatty acids revealed that Σn-3 PUFAs significantly reduced DN risk, especially EPA and DHA, while Σn-6 PUFAs did not.

Most studies have observed that higher UACR levels in patients with hypertension and insulin resistance [24,25]. A study demonstrated that hypertension significantly increased protein excretion in urine and contributed to the development of nephropathy in diabetics [26]. Because our participants were undergoing appropriate antihypertensive medication treatment, no correlation between blood pressure and UACR was observed in the present study (*p* = 0.617). The risk of developing nephropathy increases with diabetes duration, and the early diagnosis and treatment of risk factors for nephropathy can reduce the development and progression of DN [27]. Our results showed that relative to the participants with DM, those with DN took a greater variety of hypoglycemic drugs and a greater proportion of them received insulin injections; these findings suggest that the participants with DN had more difficulties controlling their blood sugar than did those with DM. Poor glycemic control in patients with DM occurs for various reasons, including inflammation response in the body, an inappropriate diet, and a lack of exercise [28,29]. Further research is required to clarify the reasons for the relatively poor glycemic control in patients with DN.

Dietary low-fat soy milk powder slows DN progression by inhibiting renal fibrosis and renal inflammation [30]. A longitudinal survey study revealed that a higher consumption of low/reduced-fat dairy foods resulted in a 49% reduction in CKD risk among older adults over the span of 10 years [26]. Another study demonstrated that the consumption of Mediterranean diet protein sources (i.e., low-fat dairy products, fish, poultry, soy products, and legumes) reduces the odds of developing DN [31]. The Atherosclerosis Risk in Communities (ARIC) study, which had a median follow-up period of 23 years, revealed that a higher dietary intake of nuts, legumes, and low-fat dairy products is associated with lower CKD risk [32]. Our results also indicated that a higher intake of fish, low-fat dairy products, soy products, and olive oil can help reduce DN risk; these foods may have a protective effect against the development of kidney disease.

Our findings indicated that consuming high-fat fish is more likely to reduce DN risk than consuming low- and moderate-fat and freshwater fish. Therefore, we further analyzed the fatty acid composition of the participants’ diets and discovered that total n-3 PUFA, especially EPA and DHA significantly reduced DN risk. Prostanoids and leukotrienes are derived from long-chain PUFAs, arachidonic acid, eicosapentaenoic acid, and docosahexaenoic acid by cyclooxygenase 1 (COX-1), cyclooxygenase 2 (COX-2) lipoxygenase 5, and cytochrome P450 pathways through distinct receptors that present at various segments of a nephron [33]. Oxygenated PUFAs are potentially involved in DN. An increasing body of evidence supports the position that COX-1 and COX-2 differentially regulate renal function. Qi et al. demonstrated that tubulointerstitial COX-2 is constitutively expressed and synthesizes natriuretic and vasodilator prostaglandins. These prostanoids maintain renal blood flow and glomerular function across a broad spectrum of effective arterial blood volume [34]. COX-2 expression and prostanoid signaling are increased in DN, and they are implicated in renal injury through the induction of inflammation and glomerular hyperfiltration, which are observed in early DN [35].

Altered intakes of dietary n-3 and n-6 PUFAs differentially influence the long-chain PUFA composition of membranes, the relative concentrations of various oxidized PUFAs, the consequent activation of signaling cascades, and, ultimately, organ physiology and pathophysiology. For example, the proinflammatory effect of arachidonic acid-derived PGE2 and leukotriene B4 is greater than that of EPA-derived PGE3 and LTB5. Long-chain n-3 PUFAs can inhibit DN progression through their anti-inflammatory effects. A hypertensive nephropathy experiment and a remnant kidney model have demonstrated that fish oil supplementation can attenuate renal injury in rats by reducing NF-κB activation, inhibiting COX-2 expression, reducing reduced nicotinamide adenine dinucleotide phosphate oxidase activation, and consequently inhibiting pro-inflammatory gene transcription and ROS formation [36,37]. Our study revealed that a higher Σn-6/Σn-3 ratio significantly increases DN risk, whereas higher n-3 PUFA, especially EPA and DHA levels significantly reduce DN risk. The type of consumed fatty acid can influence stored fatty acid composition and structural lipids in various body compartments, such as erythrocyte membranes [38]. Because of funding constraints, we could only analyze the participants’ intake and sources of dietary fatty acids and did not measure their erythrocyte fatty acid composition. Chung et al. analyzed erythrocyte fatty acids and longitudinally observed their effects on renal function decline in patients with T2DM; their results revealed that a high n-3 PUFA level or n-3/n-6 PUFA ratio has protective effects against renal function impairment [39], which is consistent with our results.

We found a higher proportion of using SGLT2 inhibitors in the DN group. Preclinical studies and clinical trials of SGLT2 inhibitors have consistently demonstrated a reduction in albuminuria and preservation of kidney function [40,41,42]. In 2018, The American Diabetes Association/European Association for the Study of Diabetes Consensus Report recommends SGLT inhibitors as the preferred add-on therapy for patients with type 2 diabetes to prevent the worsening of diabetic nephropathy [43]. Therefore, we believe that is why clinicians prescribed more SGLT2 inhibitors to patients in the DN group. The mechanisms of SGLT2 inhibitors to improve renal outcomes in patients with type 2 diabetes include the inhibition of renal glucose reabsorption, reduction in blood pressure, amelioration of glucotoxicity, and induction of hemodynamic effects [44,45]. The present cross-section study showed that the intake of n-3 PUFA might decrease the risk of DN. However, how many doses can achieve effective improvement, and the related mechanisms are still unclear. Whether n-3 PUFA can synergistically reduce DN risk with SGLT2 inhibitors also needs more research to confirm.

Through dietary surveys, our study revealed that a healthy diet that includes fish rich in n-3 PUFA, low-fat dairy products, soy products, and shellfish can help to reduce DN risk; thus, dietary advice should be provided to patients with diabetes at an early stage to prevent nephropathy. However, several limitations should be considered when interpreting our study results. First, our study was a cross-sectional study that did not explain the consequence of longitudinal efforts on DN development. Second, our study used a small sample size and all of its participants were living in northern Taiwan; these are factors that may limit the generalizability of the results. Therefore, large-sample, multicenter studies are required. Third, we did not analyze fatty acid composition and inflammatory markers. Thus, a further exploration of the mechanism of long-chain n-3 fatty acids in reducing DN risk is warranted.

## 5. Conclusions

In summary, our findings suggest that increasing the intake of high-fat marine fish, shellfish, soybean products, and low-fat dairy products can help mitigate the risk of DN and that patients with DM who consume n-3 PUFAs, high-quality proteins, are less likely to develop DN.

## Figures and Tables

**Figure 1 nutrients-14-02148-f001:**
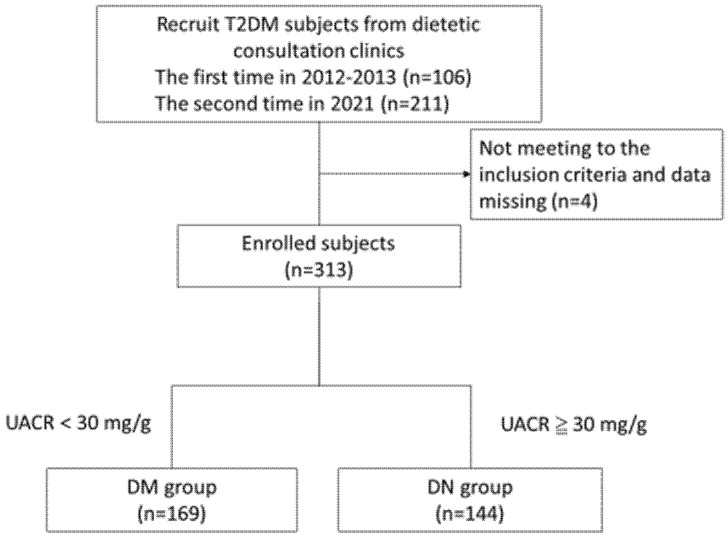
Participant recruitment and assignment process.

**Table 1 nutrients-14-02148-t001:** Demographics and clinical characteristics of participants. ^a^

Characteristic	DM ^b^ (n = 169)	DN (n = 144)	*p*-Value
Sex (M/F)	89/80	73/71	0.735
Age (years)	62.7 ± 12.5	66.0 ± 11.1	0.017 *
Diabetes duration (years)	9.3 ± 7.4	12.8 ± 8.6	<0.001 *
Body Height (cm)	161.1 ± 9.1	159.9 ± 8.2	0.201
Body Weight (kg)	68.7 ± 13.4	69.5 ± 15.6	0.614
Body mass index (kg/m^2^)	26.4 ± 4.3	27.0 ± 5.6	0.309
Waist circumference(inch)	33.7 ± 5.2	34.8 ± 5.2	0.068
Systolic blood pressure (mm Hg)	137.4 ± 13.8	136.2 ± 14.6	0.450
Diastolic blood pressure (mm Hg)	74.4 ± 9.6	74.0 ± 10.1	0.722
Use of antihypertensive drugs, n (%)			0.735
No	64 (37.9)	43 (29.9)	
One type	67 (39.6)	62 (43.1)	
Two-type combination	26 (15.4)	25 (17.4)	
More than two type combination	12 (7.1)	14 (9.7)	
Use of oral hypoglycemic drugs, n (%)			0.023 *
No	12 (11.0)	13 (13.0)	
One type	71 (65.1)	47 (47.0)	
Two-type combination	26 (23.9)	40 (40.0)	
More than two type combination	26 (23.9)	35 (35.0)	
Use of Insulin, n (%)	26 (23.9)	35 (35.0)	0.077
UACR (mg/g)	12.2 ± 8.0	349.9 ± 692.6	<0.001 *
Creatinine (mg/dL)	0.9 ± 0.3	1.2 ± 0.8	<0.001 *
eGFR (mL/min/1.73 m2)	85.6 ± 22.8	72.7 ± 30.7	<0.001 *
Fasting plasma glucose (mg/dL)	142.9 ± 46.2	153.1 ± 47.6	0.046 *
HbA1c (%)	7.3 ± 1.2	7.9 ± 1.4	<0.001 *
Triglycerides (mg/dL)	139.6 ± 98.1	164.3 ± 100.6	0.031 *
Cholesterol (mg/dL)	157.8 ± 31.6	158.6 ± 59.2	0.876

^a^ Values are presented as means ± standard deviations or n (%). ^b^ DM, diabetes mellitus; DN, diabetic nephropathy; UACR, urine albumin–creatinine ratio; eGFR, estimated glomerular filtration rate; HbA1c, Hemoglobin A1C. * Differences between DM and DN groups were tested using independent *t* tests or chi-square tests; a *p*-value < 0.05 was considered statistically significant.

**Table 2 nutrients-14-02148-t002:** Weekly consumption (no. of servings/week) of selected foods by DM and DN groups as assessed through a semiquantitative food frequency questionnaire. ^a^

Food Items	DM (n = 169)	DN (n = 144)	*p*-Value
Eggs	2.2 ± 1.5	2.1 ± 1.5	0.682
Meat and offal	7.2 ± 2.8	7.4 ± 2.7	0.527
Marine water fishes			
Low-fat	2.7 ± 2.3	2.2 ± 2.3	0.055
Moderate-fat	2.5 ± 2.4	2.1 ± 2.2	0.095
High-fat	3.0 ± 2.3	2.3 ± 2.3	0.012 *
Freshwater fishes	1.5 ± 2.0	1.7 ± 2.2	0.519
Shellfish	0.7 ± 1.2	0.5 ± 0.9	0.035 *
Processed fish products	0.3 ± 0.8	0.2 ± 0.6	0.145
Dairy products			
Whole milk	0.9 ± 1.6	0.7 ± 1.3	0.137
low-fat milk	0.3 ± 0.9	0.2 ± 0.7	0.147
skim milk	0.2 ± 0.7	0.0 ± 0.3	0.053
Soybean products	2.5 ± 2.6	1.9 ± 2.0	0.013 *
Fats and oils			
Animal fat	1.4 ± 3.1	1.6 ± 3.3	0.634
Soybean oil and sunflower oil (n-6 PUFA mainly)	5.7 ± 4.5	6.1 ± 4.6	0.431
Olive oil (n-9 MUFA mainly)	3.1 ± 3.9	2.3 ± 3.6	0.044 *
Other oils	1.4 ± 2.9	0.7 ± 2.3	0.031 *
Nuts and seeds	1.7 ± 2.7	1.2 ± 2.1	0.120

^a^ Data are presented as means ± standard deviations. * Mean significant difference between diabetes mellitus (DM) and diabetic nephropathy (DN) groups was tested through independent *t*-tests (*p* < 0.05).

**Table 3 nutrients-14-02148-t003:** Associations between consumption of selected foods and diabetic nephropathy.

Food Items	Coefficients	SE ^a^	Odds Ratio	95% CI	*p*-Value
Eggs					
Model 1	−0.026	0.077	0.975	0.837–1.135	0.742
Model 2	−0.021	0.082	0.979	0.834–1.149	0.794
Meat, Offal					
Model 1	0.026	0.042	1.027	0.946–1.114	0.525
Model 2	0.066	0.045	1.069	0.978–1.167	0.142
Marine fishes					
Low-fat					
Model 1	−0.101	0.051	0.904	0.818–0.999	0.047 *
Model 2	−0.110	0.054	0.896	0.807–0.995	0.041 *
Moderate-fat					
Model 1	−0.102	0.051	0.903	0.817–0.998	0.045 *
Model 2	−0.098	0.053	0.906	0.816–1.006	0.064
High-fat					
Model 1	−0.132	0.051	0.876	0.792–0.969	0.010 *
Model 2	−0.141	0.054	0.868	0.781–0.965	0.009 *
Freshwater fishes					
Model 1	0.037	0.055	1.038	0.933–1.155	0.493
Model 2	0.070	0.058	1.072	0.957–1.201	0.228
Shellfish					
Model 1	−0.301	0.118	0.740	0.588–0.933	0.011 *
Model 2	−0.343	0.126	0.709	0.554–0.908	0.007 *
Processed fish products					
Model 1	−0.262	0.174	0.770	0.547–1.083	0.133
Model 2	−0.354	0.190	0.702	0.483–1.018	0.062
Dairy products					
Whole milk					
Model 1	−0.095	0.082	0.910	0.774–1.069	0.249
Model 2	−0.088	0.086	0.916	0.774–1.085	0.310
Low-fat milk					
Model 1	−0.232	0.144	0.793	0.598–1.051	0.106
Model 2	−0.199	0.148	0.820	0.614–1.095	0.178
Skim milk					
Model 1	−0.497	0.264	0.608	0.362–1.021	0.060
Model 2	−0.578	0.289	0.561	0.319–0.988	0.045 *
Soybean products					
Model 1	−0.136	0.057	0.873	0.781–0.975	0.016 *
Model 2	−0.112	0.056	0.894	0.800–0.998	0.046 *
Fats/oils					
Animal fat					
Model 1	0.012	0.036	1.012	0.943–1.086	0.747
Model 2	0.003	0.038	1.003	0.931–1.081	0.933
Soybean oil and sunflower oil (n−6 PUFA mainly)					
Model 1	0.026	0.025	1.026	0.977–1.079	0.304
Model 2	0.064	0.053	1.066	0.960–1.183	0.235
Olive oil (n-9 MUFA mainly)					
Model 1	−0.074	0.030	0.928	0.875–0.985	0.014 *
Model 2	−0.060	0.032	0.942	0.884–1.003	0.062
Nuts and seeds					
Model 1	−0.057	0.049	0.945	0.859–1.040	0.244
Model 2	−0.042	0.051	0.958	0.867–1.059	0.404

^a^ SE, standard error of mean; OR, odds ratio; 95% CI, 95% confidence interval; Model 1, model with no adjustments; Model 2, model adjusted for diabetes duration, age, and HbA1c; * Mean significant difference, a *p*-value < 0.05 was considered statistically significant.

**Table 4 nutrients-14-02148-t004:** Dietary fatty acid consumption of participants. ^a^

Variables	DM ^b^ (n = 169)	DN (n = 144)	*p*-Value
Total saturated fatty acids (g)	33.8 ± 13.3	30.1 ± 13.1	0.016 *
Total monounsaturated fatty acids (g)	44.1 ± 15.7	39.6 ± 15.8	0.012 *
Total polyunsaturated fatty acids (g)	31.5 ± 10.8	28.6 ± 11.0	0.020 *
Σn-6 (g) ^b^	22.1 ± 7.3	20.8 ± 7.1	0.114
Σn-3 (g)	7.6 ± 4.3	6.4 ± 4.4	0.011 *
EPA (g)	1.9 ± 1.4	1.5 ± 1.3	0.011 *
DHA (g)	3.7 ± 2.5	3.0 ± 2.6	0.023 *
Σn-6/Σn-3 ratio	4.1 ± 3.0	4.9 ± 3.3	0.038 *

^a^ Values are presented as means ± standard deviations. ^b^ Σn-6, total amount of C18:2n-6 and C20:4n-6; Σn-3, total amount of C18:3n-3, C20:5n-3 (eicosapentaenoic acid, EPA), and C22:6n-3 (docosahexaenoic acid, DHA). * Mean significant difference between diabetes mellitus (DM) and diabetic nephropathy (DN) groups was tested through independent *t*-tests (*p* < 0.05).

**Table 5 nutrients-14-02148-t005:** Associations between dietary fatty acids and diabetic nephropathy.

Variables	β	S.E ^a^	OR	95% CI	*p*-Value
Total saturated fatty acids					
Model 1	−0.020	0.009	0.981	0.964–0.998	0.027 *
Model 2	−0.018	0.009	0.982	0.964–1.000	0.044 *
Total monounsaturated fatty acids					
Model 1	−0.017	0.007	0.983	0.969–0.998	0.023 *
Model 2	−0.015	0.008	0.985	0.970–0.999	0.042 *
Total polyunsaturated fatty acids					
Model 1	−0.022	0.011	0.978	0.958–0.999	0.039 *
Model 2	−0.020	0.011	0.981	0.960–1.022	0.079
Σn-6 ^b^					
Model 1	−0.021	0.016	0.980	0.949–1.011	0.194
Model 2	−0.014	0.017	0.986	0.954–1.018	0.390
Σn-3					
Model 1	−0.064	0.027	0.938	0.889–0.989	0.017 *
Model 2	−0.065	0.028	0.937	0.887–0.990	0.021 *
EPA					
Model 1	−0.188	0.086	0.828	0.699–0.981	0.029 *
Model 2	−0.197	0.090	0.821	0.688–0.979	0.029 *
DHA					
Model 1	−0.098	0.046	0.907	0.829–0.992	0.033 *
Model 2	−1.102	0.048	0.903	0.823–0.992	0.033 *
Σn-6/Σn-3 ratio					
Model 1	0.077	0.037	1.080	1.006–1.161	0.035 *
Model 2	0.084	0.038	1.088	1.010–1.172	0.027 *

^a^ SE, standard error of mean; OR, odds ratio; 95% CI, 95% confidence interval; EPA, eicosapentaenoic acid; DHA, docosahexaenoic acid. ^b^ Σn-6, total amount of C18:2 n-6 and C20:4 n-6; Σn-3, total amount of C18:3 n-3, C20:5 n-3 (EPA), and C22:6 n-3 (DHA). * Significant difference between diabetes mellitus (DM) and diabetic nephropathy (DN) groups was tested through binary logistic regression (*p* < 0.05).

## Data Availability

The data supporting the reported results and conclusions can be found in the submitted figure and tables. Additional research materials and protocols that are relevant to the study are available from the corresponding author upon reasonable request.

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
