# Peer review of "Association of Dietary Fish and n-3 Unsaturated Fatty Acid Consumption with Diabetic Nephropathy from a District Hospital in Northern Taiwan"

_nutrients, 2022, doi:10.3390/nu14102148_

Round 1
Reviewer 1 Report
I had a great privilege to review the manuscript entitled " Association of dietary fish and n-3 unsaturated fatty acid consumption with diabetic nephropathy from a district hospital in 3 Northern Taiwan " submitted by the authors. However, the reviewer thought this study needs to be substantially improved before it can be published.
Please see my comments below:
Major:
- Wording: (a), Please avoid words that imply causation, e.g., increase, decrease and effect. (b), it is necessary to distinguish the differences between “show” and “indicate/suggest”. When describing your results specifically, we usually use “show”; When discussing the significance of your results, we usually use “indicate/suggest”.
- Aim of this study: The current topic does not exactly match your research purpose. If you use the current stated research purpose, most of your findings will be out of statistical significance due to the multiple testing issue. Instead, you may test the association of fish and n-3 unsaturated fatty acid consumption with diabetic nephropathy as the aim of this study.
- Discussion section needs to be redesigned. You may summarize the results of the article in the first discussion paragraph.
- Have you calculated the total energy intake?
- It is better to distinguish the source of PUFA (marine omega-3 PUFAs or others)
Minor:
Line 29: what is history of diabetes? Does it mean the family history of diabetes?
Line 134, Please introduce more about the methods for calculating the PUFA and MUFA.
Line 177, Please introduce the details of covariates in the method section.
Author Response
Response to Reviewer #1:
I had a great privilege to review the manuscript entitled " Association of dietary fish and n-3 unsaturated fatty acid consumption with diabetic nephropathy from a district hospital in 3 Northern Taiwan " submitted by the authors. However, the reviewer thought this study needs to be substantially improved before it can be published.
Please see my comments below:
Major:
- Wording: (a), Please avoid words that imply causation, e.g., increase, decrease and effect. (b), it is necessary to distinguish the differences between “show’ and “indicate/suggest’. When describing your results specifically, we usually use “show’; When discussing the significance of your results, we usually use “indicate/suggest’.
Response: We thank the reviewer for the constructive suggestion. Now the words that have changed into correct wording (lines 29, 61, 273, 303 and 316) please refer the revised manuscript.
- Aim of this study: The current topic does not exactly match your research purpose. If you use the current stated research purpose, most of your findings will be out of statistical significance due to the multiple testing issue. Instead, you may test the association of fish and n-3 unsaturated fatty acid consumption with diabetic nephropathy as the aim of this study.
Response: Thank you for the comment. As suggested, the aim of study is modified to investigate the association of fish and n-3 unsaturated fatty acid consumption with diabetic nephropathy and represented at line 98-102.
- Discussion section needs to be redesigned. You may summarize the results of the article in the first discussion paragraph.
Response: Thank you for the suggestive comment. As suggested, discussion section is fixed with suggestions in revised manuscript (lines 303-308).
- Have you calculated the total energy intake?
Response: The dietary records of total energy intake per day was 1726.2 ± 334.8 kcal in DM and 1714.1 ± 776.9 kcal in DN group. There was no difference significant difference between groups. We provide the information in the revised manuscript (line 238-240).
- It is better to distinguish the source of PUFA (marine omega-3 PUFAs or others)
Response: Thank you for the helpful comment. We did classify according to the main different PUFA sources. Marine fish, freshwater fish and shellfish contain different proportions of n-3 PUFAs, so we classify these foods as different types of food. In Taiwan, the main source of n-6 PUFA is soybean oil and sunflower oil, we also classify these foods as n-6 PUFA mainly. Please see the food types in Table 2. Fats in foods are made up of a blend of fatty acids, although the proportions vary. Marine fish contains more omega-3 PUFAs, but other foods, such as meat, also contain omega-3 PUFAs, but in lower amounts. Therefore, we further calculated total consumption of various fatty acids to determine whether total n-3 PUFAs or total n-6 PUFAs were associated with the risk of DN.
Minor:
Line 29: what is history of diabetes? Does it mean the family history of diabetes?
Response: Thank you for informing us. It refers the time of diabetes duration in patients. Now it is fixed in abstract (lines 30).
Line 134, Please introduce more about the methods for calculating the PUFA and MUFA.
Response: Thank you for the helpful comment. We converted servings of each type of food into weight and analyzed fatty acid composition using the government-published Taiwan Food Nutrition Database 2020. Σn-6 PUFA was the total amount of C18:2n-6 and C20:4n-6; Σn-3 PUFA was the total amount of C18:3n-3, C20:5n-3 (eicosapentaenoic acid, EPA), and C22:6n-3 (docosahexaenoic acid, DHA); MUFA was the total amount of C18:1n-9 (added in lines 159-164).
Line 177, Please introduce the details of covariates in the method section.
Response: Thank you for the helpful comment. We added the details in line 201-202.
Reviewer 2 Report
Line 27 – instead of ‘examined’, please write ‘assessed’
Line 29 – instead of ‘content’, please write ‘intake’ as follows: ‘(…) the participants’ dietary intake’
Line 30 – ‘more likely to have a history of diabetes’ – correct to
Line 31 – replace comma after ‘DM group,’ by a full stop ‘DM group.’.
Line 31-33 – ‘Logistic regression models revealed that high fat marine fishes had the lowest odds ratio (OR) compared with other fishes (OR: 0.868; 95% CI: 0.781-0.965, p =0.009).’ – this sentence is incomplete, do you mean intake of high fat marine fishes? lowest OR for what outcome variable?
Line 37 – adjusted for what?
Line 43 – Instead of referring to DM population, refer to DM prevalence.
Line 63 – replace ‘has’ with ‘had’
Line 67/68 – not clear what is meant by ‘active blood sugar control’? Also, this sentence needs rewording, so it reads in a more coherent manner.
Line 84 – refer to the study design, was it an RCT?
Line 85 – refer to how the 4 g were consumed, via food intake or supplements? And what was the ration of n-3 PUFAS?
Line 85 – this is the first time you use n-3 polyunsaturated fatty acid so should have here (PUFA), instead of line line 93
Line 87 – refer to what type of study – cross-sectional?
Line 91 – remove ‘incident’
Line 92-95 – include study design and hazard ratio. Wording needs improving to make the sentence read better.
Line 98/99 – please write the aim in a clear manner and include all the relevant information. i.e. your study population, your outcomes (various protein sources and fatty acid compositions of what?). Please refer to the PICO framework for non-interventional studies to help formulating your aim in a clear and concise manner: https://ebm.bmj.com/content/early/2022/01/10/bmjebm-2021-111889
Line 105- need to state when recruitment started and finished.
Line 109 - justify the age range, why 30-85 and not >18 y?
Line 114 – Did participants sign consent? Need to include this information and make it clear. If not, why?
Line 122 – not clear whether data was collected from medical records or whether it was measured.
Line 131 change wording ‘lived their lives normally’
Line 166 – avoid colloquial language and use the scientific terms: mid-point between the iliac crest (hip bone) and costal margin (lowest rib).
Line 167 – add the word ‘included’ in the sentence: ‘(…) that were measured included fasting plasma glucose, (…)’
Line 168 – were all the biochemical analysis done by enzymatic colorimetric assay? Need to make it clear.
Line 178 – 180 – not clear how you divided the groups, please explain – is it DM with and without DN?
Line 192 – this information should be in the methods and a justification as to why the 9 years apart in recruitment.
Line 206 – need to include what outcomes you are referring to from the table when stating poorer glycaemic control and renal function.
Line 209/210 – please write the sentence coherently.
Table 1 – instead of ‘using’ please write ‘use’.
Table 2 – please display the food items clearly by having clear subheadings that differentiate the different grouped foods.
Line 234 - instead of ‘employed’, please write ‘used’
Line 235 – need to include the covariates in the methods section and justification for the ones used in the models. Also, need to explain the different models in the methods section.
Line 237 – be specific, lowest OR of what?
Line 281 – please correct sentence structure
Line 284 – write ‘patients with diabetes’
Be consistent with the use of n-3 PUFA, sometimes n-3 fats is used, long chain n-3 fatty acids, n-3 fatty acid
Author Response
Response to Reviewer #2:
We thank the reviewer for the constructive suggestion. We carefully reviewed the manuscript point-by-point response listed below.
Line 27 – instead of ‘examined’, please write ‘assessed’
Response: In line 27, the word was fixed.
Line 29 – instead of ‘content’, please write ‘intake’ as follows: ‘(…) the participants’ dietary intake’
Response: In lines 28, the word was fixed.
Line 30 – ‘more likely to have a history of diabetes’ – correct to
Response: In line 30, the sentence is corrected in revised manuscript.
Line 31 – replace comma after ‘DM group,’ by a full stop ‘DM group.’.
Response: In lines 31, the punctuation was fixed.
Line 31-33 – ‘Logistic regression models revealed that high fat marine fishes had the lowest odds ratio (OR) compared with other fishes (OR: 0.868; 95% CI: 0.781-0.965, p =0.009).’ – this sentence is incomplete, do you mean intake of high fat marine fishes? lowest OR for what outcome variable?
Response: In lines 31-32, the sentence was rewritten to make it more complete. ‘Logistic regression models revealed that intake of high fat marine fishes had the lowest odds ratio (OR) for DN risk compared with other fishes.’
Line 37 – adjusted for what?
Response: In lines 37, the sentence was rewritten to make it more complete. ‘regardless of whether adjusted the variables, including diabetic duration, age, and HbA1c.’
Line 43 – Instead of referring to DM population, refer to DM prevalence.
Response: In line 44, the word was fixed.
Line 63 – replace ‘has’ with ‘had’
Response: In line 63, the word was fixed.’
Line 67/68 – not clear what is meant by ‘active blood sugar control’? Also, this sentence needs rewording, so it reads in a more coherent manner.
Response: In lines 68, the sentence was rewritten to make it more complete. ‘The dietary intake is a key and modifiable factor that contribute to prevent or delay the development of DN.’
Line 84 – refer to the study design, was it an RCT?
Response: It was not a RCT. Han et al conducted a retrospective study.
Line 85 – refer to how the 4 g were consumed, via food intake or supplements? And what was the ration of n-3 PUFAS?
Response: According to Han et al study, they included 344 DM patients with a history of O3FA (highly purified ethyl ester concentrates of EPA and DHA) supplementation for managing hypertriglyceridemia. They compared the annual rates of GFR decline among three dose categories: 1 g/day, 2 g/day, and 4 g/day.
Line 85 – this is the first time you use n-3 polyunsaturated fatty acid so should have here (PUFA), instead of line line 93
Response: In line 84, the abbreviation of PUFA was used in the first time.
Line 87 – refer to what type of study – cross-sectional?
Response: Study type is added in In line 87. ‘A nested case control study’
Line 91 – remove ‘incident’
Response: In line 91, the ‘incident’ was removed.
Line 92-95 – include study design and hazard ratio. Wording needs improving to make the sentence read better.
Response: In line 92-95, the sentence was rewritten to make it more complete.’ Friedman et al. conducted a retrospective study. They measured baseline erythrocyte n-3 PUFA levels of hemodialysis patients and summed of erythrocyte EPA and DHA content as an omega-3 index (O3I). In a multivariate model, a protective trend was observed with an O3I above the median (Hazard Ratio (95% CI); 2.48 (0.88, 6.95)).’
Line 98/99 – please write the aim in a clear manner and include all the relevant information. i.e. your study population, your outcomes (various protein sources and fatty acid compositions of what?). Please refer to the PICO framework for non-interventional studies to help formulating your aim in a clear and concise manner: https://ebm.bmj.com/content/early/2022/01/10/bmjebm-2021-111889
Response: Thank you for the comment. As suggested, we added the aim and the relevant information in line 98-102.
Line 105- need to state when recruitment started and finished.
Response: The first clinical study (June 2012 to May 2013) and second clinical study (Jan. to Dec. 2021) were stated in line 118 and line 121.
Line 109 - justify the age range, why 30-85 and not >18 y?
Response: Due to the advanced age is a risk factor for worsening diabetic nephropathy. Therefore, the age range between 30 and 85 years was justified.
Line 114 – Did participants sign consent? Need to include this information and make it clear. If not, why?
Response: In line 117, the sentence was rewritten to make it more complete.
Line 122 – not clear whether data was collected from medical records or whether it was measured.
Response: In lines 125-127, the sentence was rewritten to make it more complete.
Line 131 change wording ‘lived their lives normally’
Response: In Line 134-135, the sentence was rewritten as ‘maintain a normal life’
Line 166 – avoid colloquial language and use the scientific terms: mid-point between the iliac crest (hip bone) and costal margin (lowest rib).
Response: Thank you for the helpful comment. In line 159-164, the sentence was rewritten to make it more complete.
Line 167 – add the word ‘included’ in the sentence: ‘(…) that were measured included fasting plasma glucose, (…)’
Response: We added “included” in line 174.
Line 168 – were all the biochemical analysis done by enzymatic colorimetric assay? Need to make it clear.
Response: All the biochemical analysis was done by enzymatic colorimetric assay. In line 174-176, the sentence was rewritten to make it clearer.
Line 178 – 180 – not clear how you divided the groups, please explain – is it DM with and without DN?
Response: Thank you for the helpful comment. In line 190-191, the sentence was rewritten to make it clearer. ‘Patients with diabetes who had a urinary albumin–creatinine ratio (UACR) of ≥ 30 mg/g were defined as having DN and those with UACR < 30 mg/g were defined as DM.’
Line 192 – this information should be in the methods and a justification as to why the 9 years apart in recruitment.
Response: In the first clinical trial, we did not have enough funds, so we only collected 106 cases. In 2021, we received funding from Taoyuan Armed Forces General Hospital (grant number: TYAFGH-A-110006), so we conducted a second clinical trial and collected another 211 cases. The two clinical trials have consistent results, so we combined the data for statistical analysis. This information was added in line 185-189.
Line 206 – need to include what outcomes you are referring to from the table when stating poorer glycaemic control and renal function.
Response: In line 218-222, the sentence was rewritten to make it more clear and complete.
Line 209/210 – please write the sentence coherently.
Response: In line 218-229, the sentence was rewritten to make it more clear and complete.
Table 1 – instead of ‘using’ please write ‘use’.
Response: The words that have changed into correct wording in Table 1.
Table 2 – please display the food items clearly by having clear subheadings that differentiate the different grouped foods.
Response: Thank you for informing us. Now it is corrected in Table 2.
Line 234 - instead of ‘employed’, please write ‘used’
Response: In line 256, the word was fixed.
Line 235 – need to include the covariates in the methods section and justification for the ones used in the models. Also, need to explain the different models in the methods section.
Response: The details of covariates adjustment was added in the method section in line 201-202.
Line 237 – be specific, lowest OR of what?
Response: In line 258-259, the sentence was rewritten to make it clearer. ‘Our data revealed that high fat marine fish intake had the lowest odds ratio (OR) of DN risk compared with other fishes.’
Line 281 – please correct sentence structure
Response: In line 272-273, the sentence was rewritten to make it more complete.
Line 284 – write ‘patients with diabetes’
Response: Sorry I couldn't find it to reply.
Be consistent with the use of n-3 PUFA, sometimes n-3 fats is used, long chain n-3 fatty acids, n-3 fatty acid
Response: Thank you so much for the helpful remark. Now the words that have changed into correct wording (lines 337, 355 and 371) please refer the revised manuscript.
Round 2
Reviewer 1 Report
The response looks good, I don't have other comments,
Author Response
We thank the reviewer for the constructive comments and suggestions.